# A cryo-electron microscopy support film formed by 2D crystals of hydrophobin HFBI

Hongcheng Fan[1,2], Bo Wang[3], Yan Zhang[1], Yun Zhu [1], Bo Song[3], Haijin Xu[3], Yujia Zhai[1], Mingqiang Qiao [3,4 ✉] & Fei Sun [1,2,5,6,7 ✉]

Cryo-electron microscopy (cryo-EM) has become a powerful tool to resolve high-resolution structures of biomacromolecules in solution. However, air-water interface induced preferred orientations, dissociation or denaturation of biomacromolecules during cryo-vitrification remains a limiting factor for many specimens. To solve this bottleneck, we developed a cryo-EM support film using 2D crystals of hydrophobin HFBI. The hydrophilic side of the HFBI film adsorbs protein particles via electrostatic interactions and sequesters them from the air-water interface, allowing the formation of sufficiently thin ice for high-quality data collection. The particle orientation distribution can be regulated by adjusting the buffer pH. Using this support, we determined the cryo-EM structures of catalase (2.29 Å) and influenza hae-magglutinin trimer (2.56 Å), which exhibited strong preferred orientations using a conventional cryo-vitrification protocol. We further show that the HFBI film is suitable to obtain high-resolution structures of small proteins, including aldolase (150 kDa, 3.28 Å) and haemoglobin (64 kDa, 3.6 Å). Our work suggests that HFBI films may have broad future applications in increasing the success rate and efficiency of cryo-EM.

[1] National Key Laboratory of Biomacromolecules, CAS Center for Excellence in Biomacromolecules, Institute of Biophysics, Chinese Academy of Sciences, 100101, Beijing, China. [2] University of Chinese Academy of Sciences, Beijing, China. [3] The Key Laboratory of Molecular Microbiology and Technology, Ministry of Education, College of Life Sciences, Nankai University, 300071, Tianjin, China. [4] School of Life Science, Shanxi University, Shanxi, China. [5] Center for Biological Imaging, Institute of Biophysics, Chinese Academy of Sciences, 100101, Beijing, China. [6] Physical Science Laboratory, Huairou National Comprehensive Science Center, No. 5 Yanqi East Second Street, 101400, Beijing, China. [7] Bioland Laboratory, 510005, Guangzhou, Guangdong Province, China. ✉email: qiaomq@nankai.edu.cn; feisun@ibp.ac.cn

With recent breakthroughs and developments, cryo-electron microscopy (cryo-EM) has become a powerful technique to study the high-resolution three-dimensional structures of biological macromolecules in solution[1]. Very recent reports demonstrated that cryo-EM can even reach atomic resolution[2,3]. However, there are still bottlenecks to overcome for a greater success rate and higher efficiency. One is how to routinely prepare suitable cryo-vitrified samples for high-resolution and high-quality data collection[4].

The cryo-vitrification procedure using the conventional holey carbon support film presents many challenges, including protein denaturation/degradation and preferred orientation induced by the air–water interface[5], nonuniform/poor distribution of protein particles in the holes[6,7], poorly controlled ice thickness, beam-induced motion, and background noise, which have become important barriers to high-resolution structure determination for many biomacromolecules by cryo-EM.

Past efforts have been made to solve these challenges. One approach is changing the surface properties of the holey carbon support foil, e.g., by glow discharging[8] or by special treatment with surfactants or PEG[9]. This approach can improve the particle distribution in the hole; however, the screening procedures were specimen dependent and exhausted without clear clues and confidence of success. A multiple blotting approach was used to increase the number of particles in the hole[7]. Holey metal support foils, including gold foil[10,11] and amorphous nickel–titanium alloy foil (ANTA foil)[12], were invented for use instead of carbon foil to reduce beam-induced motion by increasing the conductance of the support foil and to improve the quality of the cryo-vitrified specimen by decreasing nonspecific interactions between particles and foils. However, these approaches did not solve the air–water interface challenge.

Another approach is adjoining a layer of continuous film, e.g., ultrathin carbon[13], graphene[14–20] or graphene derivates[21–28], lipid monolayers[29,30] or 2D crystals of streptavidin[31,32], on top of the holey support foil. This approach can increase the density of particles in the hole, sequester the specimen from the air–water interface, and alter the particle distribution. However, ultrathin carbon films[13], PEGylated graphene oxide[24] or lipid monolayers[29,30] will generate nonnegligible noisy background, which limits the application to small particles. The preparation of a streptavidin monolayer-coated grid is sophisticated, and the additional lipid monolayer and ultrathin carbon film also generate a nonnegligible noisy background. In addition, the potential nonhydrophilic surface of carbon film or graphene materials would induce another significantly preferred orientation of adsorbed particles.

In addition, to address the air–water interface and nonuniform particle distribution problems, several instruments have been invented, such as Spotiton[33] and VitroJet[34], in an effort to minimize the time of the vitrification procedure to reduce the chance of particles attaching to the air–water interface. However, a recent study suggested that the air–water interface problem could not be effectively solved even when using a time-resolved vitrification device that could freeze samples within 6 ms[35].

Hydrophobins (HFBs) are a family of low-molecular-weight proteins (7–15 kDa) found in filamentous fungi[36]. HFBs contribute to the amphipathic film on the surface of fungal aerial hyphae and spores and play a key role in different physiological phases of fungi (Fig. 1a). HFBs exhibit a compact and globular structure that is stabilized by four conserved intramolecular disulfide bonds[37]. Interestingly, HFBs are amphiphilic proteins that display distinct hydrophobic and hydrophilic patches on the surface, making them one of the most surface-active protein families. HFBs can self-assemble into an amphipathic layer that is tightly adsorbed at a hydrophilic–hydrophobic interface, such as the air–water interface or water–solid interfaces[38]. These unique properties of HFBs have led to many biotechical applications, including emulsions, drug delivery, tissue engineering[39], biosensors[40,41], and protein purification/immobilization[42,43]. According to the hydropathy patterns and the properties of the self-assembled films, HFBs are categorized into class I and class II[44,45]. Class I HFBs contain a long, disordered, and non-conserved loop (~24 residues) between the third and fourth cysteine, while this loop is absent in class II HFBs. Class I HFBs normally self-assemble and form rodlet nanostructures, which are robust and disintegrate only in pure trifluoroacetic acid or formic acid[46,47]. Class II HFBs can self-assemble into none-rodlets nanostructures, which are sensitive to 2% SDS and 60% ethanol[48,49]. The amphiphilic hydrophobin HFBI (7.5 kDa) isolated from *Trichoderma reesei* is a class II hydrophobin known to adhere to the air–water interface without denaturation and self-assemble into a crystalline monolayer film[50]. This self-assembled 2D crystal film at the air–water interface not only lowers the surface tension of the water but also has a very high surface shear elasticity that can stabilize air bubbles and foams[51].

In this study, we explored the potential application of HFBs to solve the air–water interface challenge in the cryo-EM field. We selected the 2D crystal of HFBI as a new support film (HFBI film) for the cryo-vitrification procedure. We find that the HFBI film is intrinsically hydrophilic with low background and adsorbs protein particles via electrostatic interactions. By testing six different protein samples, apoferritin (480 kDa), glutamate dehydrogenase (GDH) (334 kDa), catalase (240 kDa), influenza haemagglutinin (HA) trimer (190 kDa), aldolase (150 kDa) and haemoglobin (64 kDa), we show that the HFBI film can protect the specimen away from the air–water interface, solve the conventional severe particle orientation preference problem, improve the quality of the cryo-EM micrograph by forming thin enough ice, and produce high-resolution structures of small proteins with molecular weights as low as 48 kDa.

## Results

**Preparation of cryo-EM grid covering by HFBI film.** We expressed recombinant HFBI in *Pichia pastoris* and purified the protein by ultrafiltration and acetonitrile extraction (Supplementary Fig. 1a, b) as described in our previous study[52]. Consistently, we found that HFBI can self-assemble into a uniform monolayer (HFBI film) with a thickness of ~1.4 nm and a root mean square roughness of $0.2 \pm 0.02$ nm (Supplementary Fig. 1c). We also found that the HFBI monolayer can alter the hydrophobic surface of the siliconized glass to hydrophilic (Supplementary Fig. 1d) and alter the hydrophilic surface of mica to relatively hydrophobic (Supplementary Fig. 1e) according to the water contact angle measurement, suggesting that the HFBI film is amphipathic.

We transferred the HFBI film to a holey amorphous nickel–titanium alloy foil grid (ANTA foil grid)[12]. After the large, visible HFBI film spontaneously formed on top of a solution drop, we incubated the ANTA foil grid on top of the drop with the ANTA foil in contact with the hydrophobic side of the HFBI film, resulting in the transfer of the HFBI film onto the grid (Fig. 1b). Since the other exposed side of the HFBI film is hydrophilic (Fig. 1c, d), conventional grid treatments by plasma cleaning or ultraviolet (UV) irradiation before cryo-vitrification are not necessary for the HFBI film. The production procedure of HFBI film-coated grids is simple and highly reproducible. No specific storage conditions are needed for the air-dried HFBI film-coated grids, which are simply stored in the grid box at room temperature. The grids can be stored for one week with no change in performance.

We then examined the integrity of the HFBI film by using scanning electron microscopy (SEM) and found that most areas (> 90%) of the holey ANTA foil were uniformly covered by the HFBI

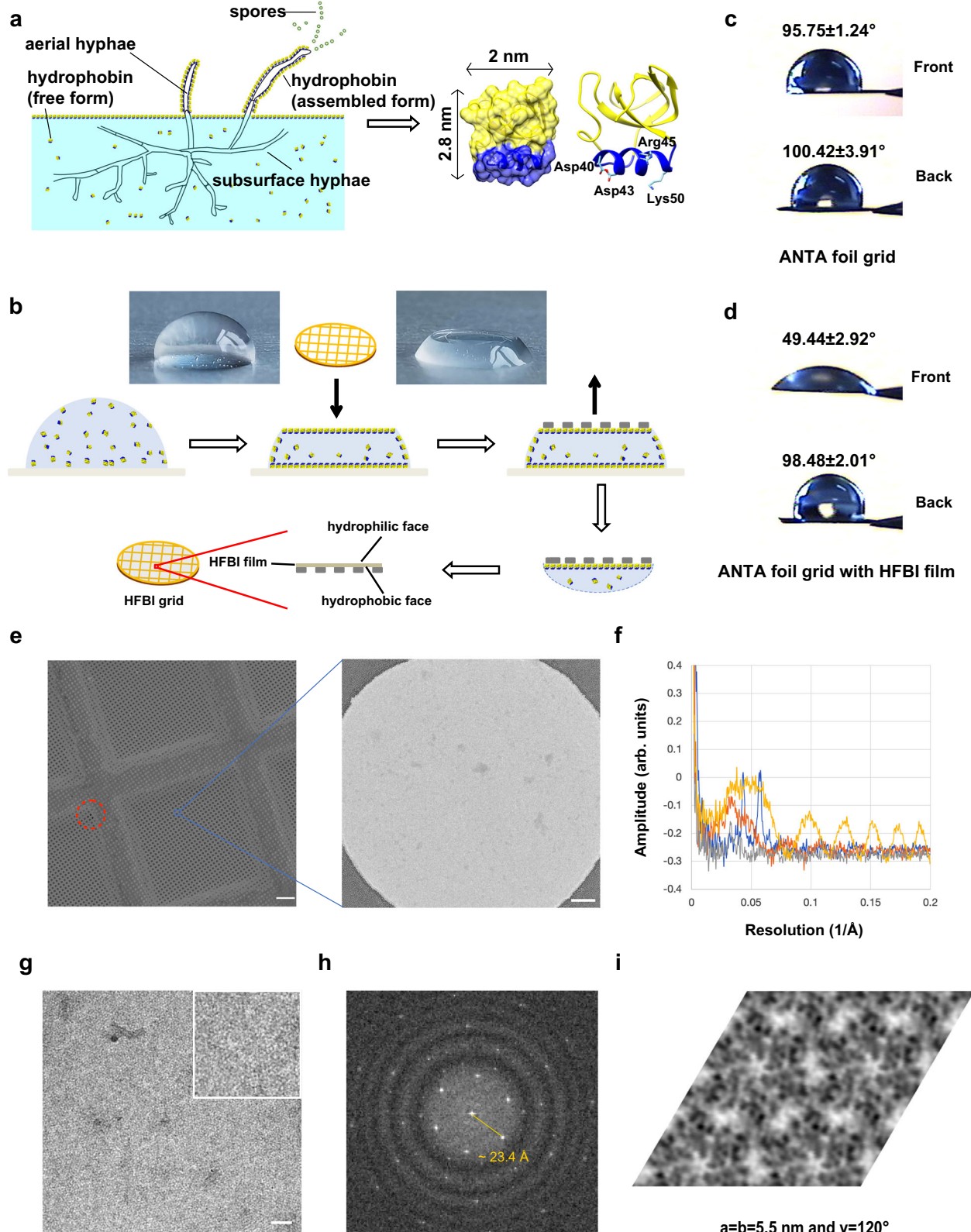

film (Fig. 1e; Supplementary Fig. 2). Based on transmission electron micrographs, we performed a quantitative analysis of contrast loss using HFBI film compared with monolayer graphene oxide film and continuous ultrathin carbon film (~2.7 nm thickness) by computing their power spectra (Fig. 1f). The monolayer graphene oxide film shows a background signal only in the low-resolution region, while the air-dried HFBI film contributes a background signal mainly in

two narrow low-resolution bands. Both are much weaker than the background generated from the continuous ultrathin carbon film.

In the subsequent cryo-EM experiment using HFBI film with vitrified solution, we observed the crystal lattice of the HFBI film at high magnification (Fig. 1g), and the power spectrum showed the first diffraction lattice spots at ~23.4 Å (Fig. 1h). We then performed further image processing using the *2dx* software

**Fig. 1 Production and characterization of HFBI film. a** The biological nature of hydrophobin HFBI and its physiological role during filamentous fungal growth[36,37]. HFBIs are secreted as monomers and assemble at the air–water interface, where they coat surfaces of aerial hyphae and spores to protect from air-drying. **b** Scheme of preparing an HFBI film-covered ANTA foil grid. **c** Water contact angles of the front and back sides of the ANTA foil grid. **d** Water contact angles of the front and back sides of HFBI film-covered grid. The HFBI film was covered on the front side. No plasma cleaning treatment was performed and 3 μl drops of water were used for the measurements in (**c**) and (**d**). **e** SEM image (left; scale bar, 10 μm) of the HFBI film-covered ANTA foil grid with a magnified TEM image (right; scale bar, 200 nm) of the HFBI film. Most regions are covered by HFBI film while a broken area is marked by the red dashed circle. **f** Power spectral profiles of TEM micrographs of different support films (amorphous carbon film in yellow, monolayer graphene oxide film in orange and HFBI film in blue) and vacuum (grey). All micrographs were acquired using FEI Titan Krios (300 kV) and K2 camera in super-resolution mode with a physical pixel size of 0.65 Å and a total dose of 25 $e^-/Å^2$. All profiles were computed using EMAN2.3[87], normalized to the total image intensity and plotted using Microsoft Excel. The thickness of the amorphous carbon film was 2.7 nm estimated by AFM. The amplitude is shown in an arbitrary unit. **g** Cryo-EM micrograph of the HFBI film with protein buffer blotted. Scale bar, 20 nm. **h** Zoomed-in power spectrum of the cryo-EM micrograph, showing the typical diffraction pattern of the HFBI film with the first diffraction lattice spots at ~23.4 Å. **i** The crystal lattice image of the HFBI film with a symmetry of p3 was obtained using the 2dx package[53] with unbending and Fourier extraction. The unit cell parameter of the lattice is labelled.

package[53] and found that the crystal lattice of the HFBI film is hexagonal with p3 symmetry (Fig. 1i; Supplementary Fig. 1f), which is consistent with the results of a previous study performed by atomic force electron microscopy[54].

**HFBI film allows sufficiently thin ice and well-distributed particles**. We first selected human apoferritin as a typical sample to explore the suitability of the HFBI film for cryo-EM single-particle analysis. We performed a conventional cryo-vitrification procedure using an FEI Vitrobot Mark IV. Using the HFBI film-covered ANTA foil grid, we collected a cryo-EM dataset on human apoferritin by using an FEI Talos Arctica operated at 200 kV and equipped with a GATAN BioQuantum K2 camera (Supplementary Table 1).

We found that after vitrification, the HFBI film could still maintain a high coverage rate and enable the formation of uniform thin ice containing apoferritin particles (Fig. 2a–c). We took a series of electron micrographs using various defocuses from −1.92 to −0.48 μm and found that the apoferritin particles could show good contrast even at a small defocus of −0.48 μm (Fig. 2c; Supplementary Fig. 3a), implying that the ice formed was sufficiently thin. We further quantitatively measured the ice thickness based on the inelastic mean free path of electrons[55] and found that the average ice thickness was ~33 ± 10 nm, which was thinner than the value of 45 ± 12 nm obtained in the control experiment without HFBI film (Supplementary Fig. 3b, c).

In addition, by inspecting the power spectrum of cryo-EM micrographs, we observed that 99.3% of the HFBI film was crystallized (Fig. 2d) and 93.8% was polycrystalline (Supplementary Fig. 3a, d). This observation is consistent with a previous report that the size of HFBI monocrystals was normally < 2 μm[50].

We also found that the apoferritin particles showed a uniformly dispersed, low-aggregation distribution, suggesting a high-quality cryo-vitrified specimen. Our subsequent cryo-electron tomographic reconstruction indicated that almost all the particles within the ice formed a single thin layer and were adsorbed onto the HFBI film, preventing air–water interface contact (Supplementary Fig. 4).

After several steps of single-particle analysis (Supplementary Figs. 5 and 6a, b), we obtained the final 3D cryo-EM map of human apoferritin (Fig. 2e) at an overall resolution of 1.96 Å according to the gold standard Fourier shell correlation threshold $FSC_{0.143}$ (Fig. 2f; Supplementary Fig. 6c). We analysed the apoferritin particle orientation distribution and found a nearly even distribution with the calculated cryo-EF[56] value of 0.86 (Supplementary Fig. 6b,d), which is better than that (cryo-EF = 0.7) of a previous report using a graphene monolayer-covered grid[16].

At a resolution of 1.96 Å, we could clearly see holes in aromatic residues such as phenylalanine and tyrosine (Supplementary Fig. 6g). We further performed density modification[57] to improve

the quality of the map (Supplementary Fig. 6h), which is comparable to the recently reported cryo-EM map of apoferritin at a resolution of 1.75 Å (Supplementary Fig. 6e, f), the highest resolution reached by using a 200 kV cryo-electron microscope[58].

We estimated the Rosenthal–Henderson $B$-factor[59] of this dataset as 87.8 $Å^2$ (Fig. 2g), suggesting that we could reach sub-3 Å resolution with only 200–400 particles of this dataset. We noted that the $B$-factor of the 1.75 Å apoferritin map[58] is 90.7 $Å^2$. We believe that the sufficiently thin ice of the cryo-vitrified specimen is responsible for the high quality and small $B$-factor of the dataset.

**HFBI film solves the strong effect of preferential orientation**. It has been shown that the air–water interface using the conventional cryo-EM grid induces 90% of particles to attach to the air–water interface, resulting in a significant preferential orientation of particles in many cases[60]. Our above study of the HFBI film using human apoferritin provided the insight that the HFBI film could protect the protein particles from the air–water interface and thus solve the associated strong effect of preferential orientation.

In the first case, we selected catalase for testing (Supplementary Table 1). Although the crystal structure of catalase was previously available, the structure of catalase had not been solved successfully by cryo-EM. Using the conventional cryo-EM sample preparation protocol, catalase exhibited a strong preferred orientation with its side view[61] (see also Fig. 3a, b), which made it impossible to solve its structure at high resolution. However, after cryo-vitrification of catalase using the HFBI film-covered ANTA foil grid, we observed a greatly improved distribution of catalase particles with various views, which was further assessed by 2D classification (Fig. 3c). After several steps of single-particle analysis (Supplementary Fig. 7), we obtained the cryo-EM map of catalase at a resolution of 2.29 Å (Fig. 3d; Supplementary Fig. 8a) according to the gold standard threshold $FSC_{0.143}$ (Fig. 3e). The corresponding Rosenthal–Henderson $B$-factor is 79.3 $Å^2$ (Supplementary Fig. 8b), which is small and indicates the high quality of the dataset.

We further analysed the particle orientation distribution and observed a nearly even distribution of catalase particles (Supplementary Fig. 8c) with a calculated cryo-EF value of 0.80 (Fig. 3c), which is significantly improved in comparison with the cryo-EF values of 0.24 (thin ice, Fig. 3a) and 0.59 (thick ice, Fig. 3b) in our control experiments using ANTA foil grids without HFBI film coating. We noted that a previous study using a holey pure gold foil grid reported a small cryo-EF value of 0.2[61].

In the second case, we selected the influenza HA trimer as another specimen for testing (Supplementary Table 1). The pursuit of a very high-resolution structure of the influenza HA trimer by cryo-EM has suffered greatly from the strong

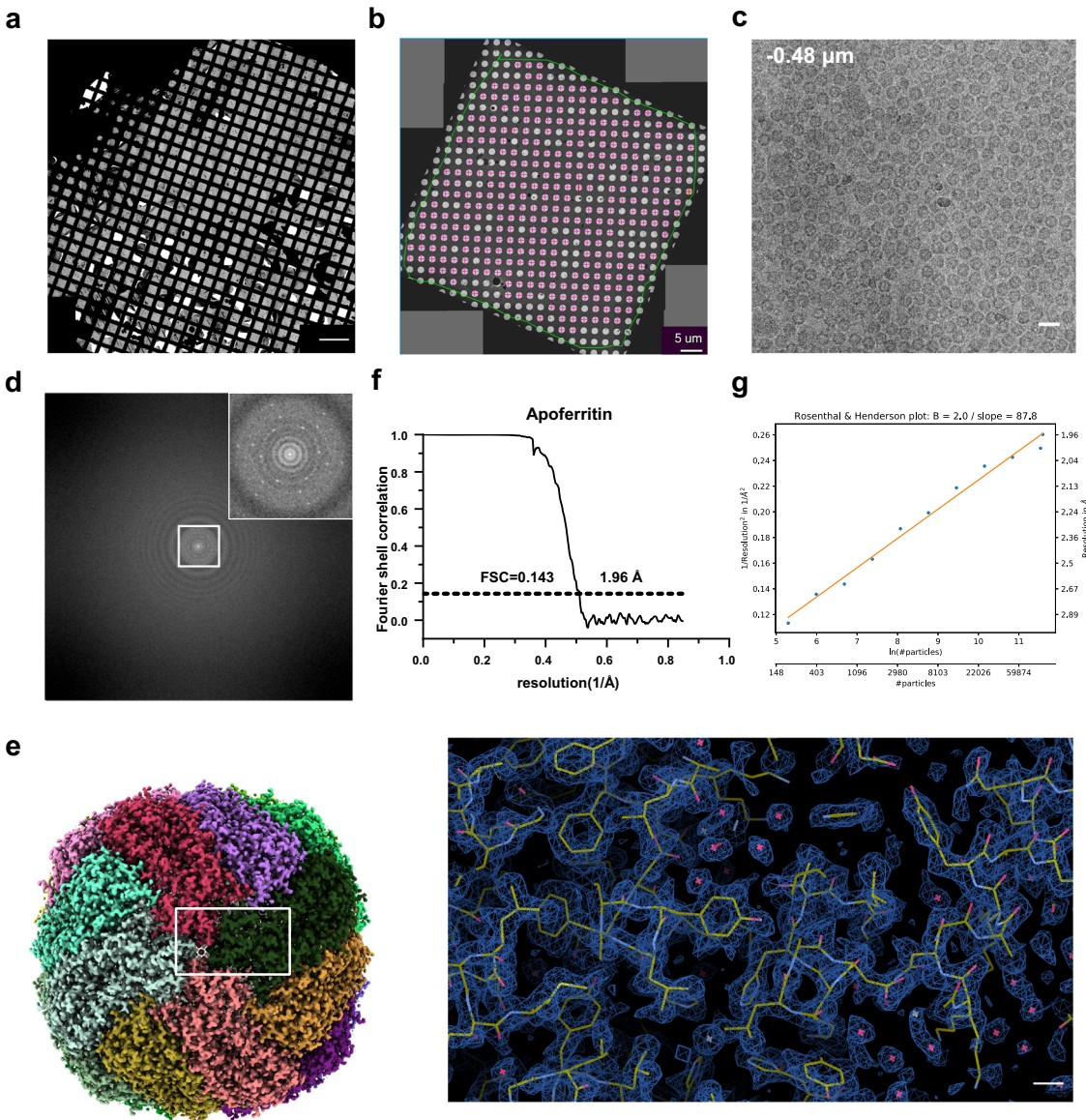

**Fig. 2 Cryo-EM application of HFBI film using human apoferritin as a test sample. a** Cryo-EM montage micrograph (scale bar, 200 μm) of the HFBI film-covered ANTA foil grid with vitrified human apoferritin. **b** Cryo-EM montage micrograph (scale bar, 5 μm) of one square of the grid. The holes with good ice thickness and relatively low contamination for data collection are labelled in purple. **c** Representative cryo-EM micrograph of human apoferritin particles. Scale bar, 20 nm. **d** Power spectrum of the high magnification cryo-EM micrograph in (**c**). The central region is magnified (inset) and shows the diffraction spots of the HFBI film. **e** Cryo-EM map of human apoferritin at 1.96 Å resolution, and a representative region is magnified (right) with the fitted atomic model. Scale bar, 2 Å. **f** Gold standard FSC curve showing an overall resolution of 1.96 Å. **g** Rosenthal–Henderson plot with an estimated *B*-factor of 87.8 Å.

preferential orientation effect with a majority top-view orientation, which was believed to be induced by the air–water interface[62]. Previous effects to solve this problem included a series of specimen tilting[62] and fast freezing processes using SPOTITION[33] or Back-it-up (BIU)[63] approaches. Here, we utilized the HFBI film-covered ANTA foil grid to prepare the cryo-vitrified HA trimer and observed a large portion of tilt and side views from raw cryo-EM micrographs (Supplementary Fig. 9a), which was further assessed by 2D classification (Fig. 4a). After several steps of single particle analysis (Supplementary Fig. 10), we obtained the cryo-EM map of the HA trimer (Supplementary Fig. 9b, c) at a resolution of 2.6 Å according to the gold standard threshold FSC$_{0.143}$ (Fig. 4b). The corresponding Rosenthal–Henderson *B*-factor is 185.6 Å$^2$ (Supplementary Fig. 9d).

We further analysed the particle orientation distribution and observed a nearly even distribution with the calculated cryo-EF value of 0.82 (Supplementary Fig. 9b), eliminating the previous strong preferential orientation effect using the conventional cryo-EM workflow. Compared to previous solutions, including specimen tilting[62,64], SPOTITON[33], and BIU[63], our method using the HFBI film can achieve the highest resolution with a comparable number of particles and the best isotropic resolution according to the criteria of 3D-FSC sphericity[62] (Fig. 4c; Supplementary Fig. 9e).

**Electrostatic interaction between HFBI film and protein particles.** Considering the hydrophilic nature of the exposed side of the HFBI film, we believe that the protein particles are adsorbed onto the HFBI film mainly via electrostatic interactions[65]. To test

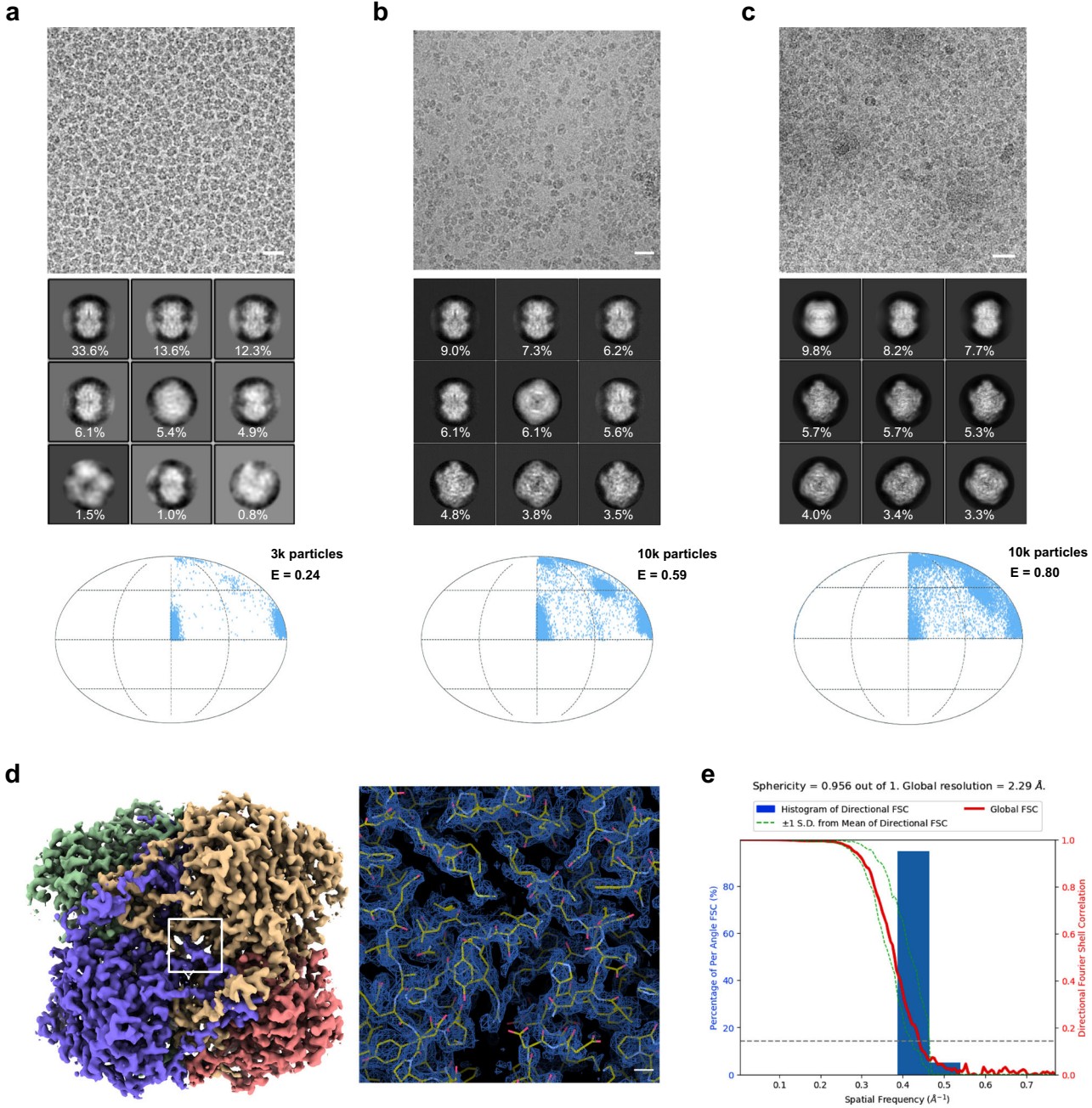

**Fig. 3 Using HFBI film to solve the preferred orientation problem of catalase. a–c** Representative cryo-EM micrographs (scale bar, 20 nm) of catalase vitrified using the ANTA foil grid in thin ice (**a**), in thicker ice (**b**) and vitrified using the HFBI film covered ANTA foil grid (**c**). The corresponding representative 2D class averages of catalase are shown in the middle. The percentage of the number of particles in each class is labelled accordingly. The computed orientation distributions with the cryo-EF values are shown at the bottom. The number of particles used to calculate the cryo-EF values is indicated accordingly. **d** Cryo-EM map of catalase at a resolution of 2.29 Å that was solved using the HFBI film-covered ANTA foil grid. A representative region of the cryo-EM map is magnified (right) with the fitted atomic model. Scale bar, 2 Å. **e** 3D-FSC plot of cryo-EM reconstruction of catalase using the HFBI film covered ANTA foil grid.

this hypothesis, we studied the orientation distribution of GDH in a cryo-vitrified specimen using the HFBI film and explored whether its distribution can be regulated by pH.

We found that the orientation distributions of GDH changed at different pH conditions from 5.5 to 8.5 (Fig. 5a; Supplementary Fig. 11a–g). For specimens with pH values of 5.5, 6.0, 6.5, 7.0, 7.5, 8.0 and 8.5 vitrified using HFBI film-coated ANTA foil grids, 240,067, 198,712, 228,000, 241,074, 223,500, 216,510 and 527,163 particles were automatically picked. Multiple rounds of 2D classification were performed to discard contaminations and

wrong particles. Finally 162,042 (48,499 for side views), 120,274 (14,013 for side views), 149,320 (37,812 for side views), 120,149 (73,256 for side views), 82,340 (52,545 for side views), 88,585 (60,808 for side views), 174,836 (126,218 for side views) particles were selected. Under acidic conditions, GDH particles mainly showed top ("triangle-like" appearance) and tilt views, while under basic conditions, GDH particles mainly showed side views ("worm-like" appearance).

To exclude the possibility that our observed pH-dependent change in orientation distribution might be induced by pH-dependent

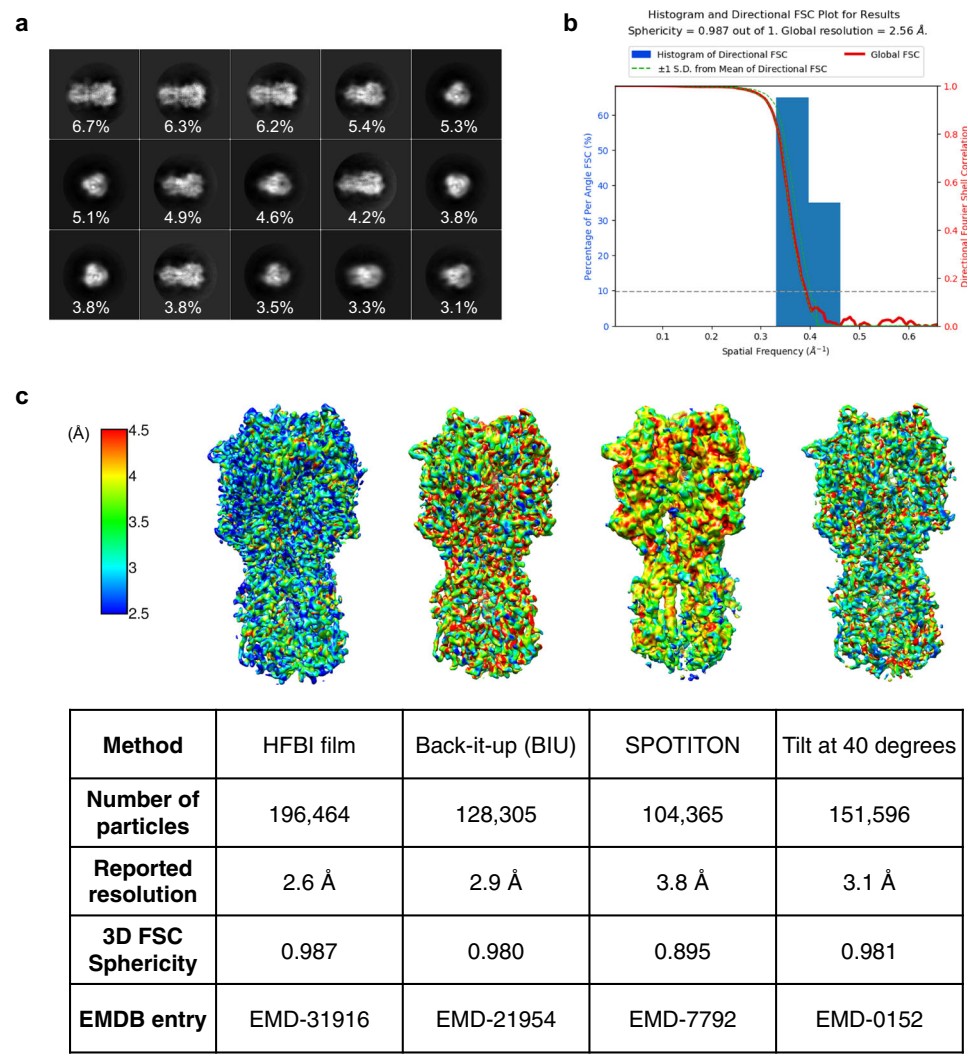

| Method | HFBI film | Back-it-up (BIU) | SPOTITON | Tilt at 40 degrees |
|---|---|---|---|---|
| **Number of particles** | 196,464 | 128,305 | 104,365 | 151,596 |
| **Reported resolution** | 2.6 Å | 2.9 Å | 3.8 Å | 3.1 Å |
| **3D FSC Sphericity** | 0.987 | 0.980 | 0.895 | 0.981 |
| **EMDB entry** | EMD-31916 | EMD-21954 | EMD-7792 | EMD-0152 |

**Fig. 4 Using HFBI film to solve the preferred orientation problem of the influenza HA trimer. a** Representative 2D class averages of the influenza HA trimer from the cryo-EM dataset collected using the HFBI film-covered ANTA foil grid. Both side and top views can be captured. The percentage of the number of particles in each class is labelled accordingly. **b** 3D-FSC plot of the cryo-EM reconstruction of influenza HA trimer using the HFBI film covered ANTA foil grid. **c** Comparisons of influenza HA trimer cryo-EM maps solved using different sample preparation approaches. "Tilt 40 degrees", the specimen was vitrified using a normal Quantifol Au grid, and the grid was tilted to 40° during data collection to solve the preferred orientation[62,64]. "SPOTITON", the specimen was fast vitrified using a self-wicked grid and Spotiton device[33]. "Back-it-up (BIU)", the specimen was rapidly vitrified by combining ultrasonic application and a through-grid wicking approach using a glass fibre filter[63]. "HBI-film", the specimen was vitrified using the HFBI film covered ANTA foil grid (this work). The cryo-EM maps of the influenza HA trimer are coloured according to their local resolutions (from 4.5 Å in red to 2.5 Å in blue), which were estimated using DeepRes[88]. The number of particles used for the final reconstructions, reported resolution, 3D-FSC sphericity and the corresponding data entry (EMDB) are shown for each approach.

characteristics of GDH itself, we performed control cryo-EM experiments using normal ANTA foil grids (Fig. 5a; Supplementary Fig. 11a–g). For specimens with pH values of 5.5, 6.0, 6.5, 7.0, 7.5, 8.0 and 8.5 vitrified using ANTA foil grids, 300,185, 255,171, 243,678, 265,459, 257,553, 235,488 and 267,823 particles were automatically picked. Finally 183,707 (71,119 for side views), 137,079 (32,461 for side views), 96,023 (15,064 for side views), 138,633 (43,985 for side views), 169,991 (34,426 for side views), 150,087 (34,041 for side views), 239,545 (73,569 for side views) particles were selected. We found that without the support of HFBI film, GDH always shows a dominant top view regardless of the pH conditions, which is significantly different from our observation using the support of HFBI film. Thus, our observed pH-dependent changes in the GDH orientation distribution are related to the pH-dependent interaction between GDH particles and the HFBI film.

With HFBI film as a support, we collected a large cryo-EM dataset of GDH at pH 7.5 (Supplementary Table 1), which is close to its isoelectric point (pH 7.66). At this pH, GDH particles exhibited the most even distribution (Supplementary Fig. 11e). After several steps of single-particle analysis (Supplementary Fig. 12), we obtained the cryo-EM map of GDH at a resolution of 2.26 Å (Fig. 5b; Supplementary Fig. 13a, b) according to the gold standard threshold $FSC_{0.143}$ (Fig. 5c). The cryo-EF analysis of this dataset gave a factor of 0.84 (Supplementary Fig. 13c), indicating no strong preferential orientation problem (Supplementary Fig. 13d). We also calculated the Rosenthal–Henderson $B$-factor of this dataset as 76.0 Å$^2$ (Supplementary Fig. 13e), again suggesting a high-quality cryo-EM dataset.

Based on the structure of GDH, we were able to analyse the electrostatic potentials of its contact sites on the HFBI film, which were plotted against the change in pH (Fig. 5d). We found that the contact site in the top view is highly positively charged at pH 5.5 and becomes less charged with increasing pH, whereas the contact site in the side view is less charged at pH 5.5 but becomes highly negatively charged at pH 8.5. At the same time, the

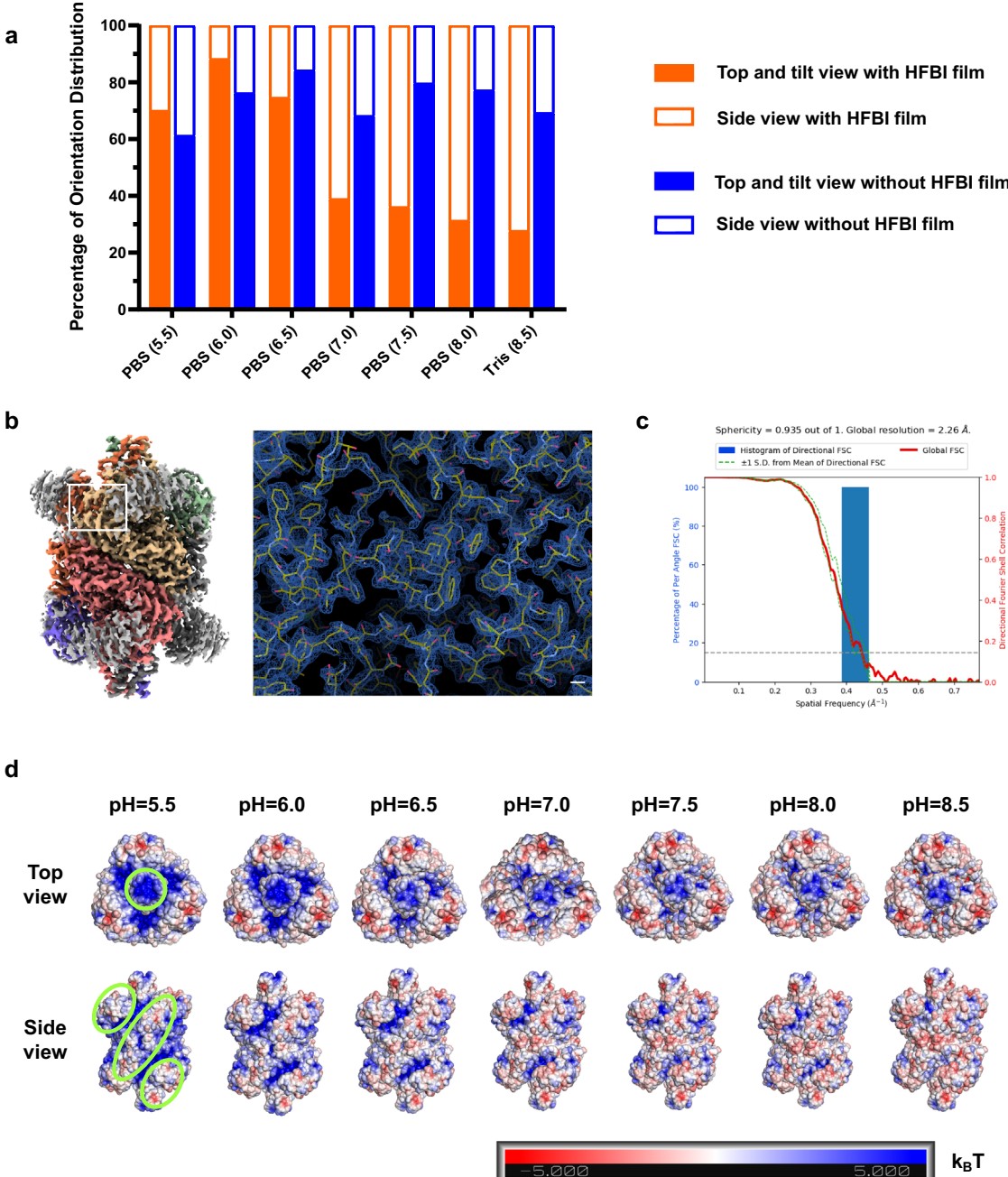

**Fig. 5 Regulating particle orientation distribution by changing the buffer condition. a** The statistics of percentages of different GDH views in different buffer conditions. PBS phosphate-buffered saline. Tris, Tris buffer (20 mM Tris–HCl and 150 mM NaCl). The pH value of each buffer is shown in brackets. **b** The 2.26 Å resolution cryo-EM map of pH 7.5 buffered GDH that was solved using the HFBI film covered ANTA foil grid, and a representative region of the cryo-EM map is magnified (right) with the fitted atomic model. Scale bar, 2 Å. **c** 3D-FSC plot of cryo-EM reconstruction of pH 7.5 buffered GDH using the HFBI film covered ANTA foil grid. **d** The electrostatic potentials of GDH at different pH values are mapped onto its solvent-accessible surface. The electrostatic potentials are coloured from red (negative charge) to blue (positive charge) in the range of −5.0 to 5.0$k_{B}T$. Green circles highlight the regions attached to the HFBI film at the corresponding views.

adsorption surface of the HFBI film contains both negative and positive charges, and its electrostatic potential is not sensitive in the pH range of 5.5–8.5 (Supplementary Fig. 14). These analyses are highly correlated with our observed pH-dependent change in GDH orientation distribution (Fig. 5a), suggesting that the HFBI film adsorbs protein particles mainly via electrostatic interactions.

The "worm-like" structure of GDH (Supplementary Fig. 11a–g) suggests an intrinsic top-to-top interaction between GDH particles. At the condition without HFBI film, the interaction between air–water interface and the top side of GDH is stronger than the top-to-top interaction itself, which enforces the majority of GDH top views. At the condition with HFBI film, the interaction between the new interface and the top side of GDH is regulated efficiently by pH and becomes weaker than the top-to-top interaction of GDH when pH is larger than 7.0, which allows the significant population of side views. Indeed, the "worm-like" structures were also observed from the recent report when the property of air–water interface is changed by adding the amount of cationic detergent[66].

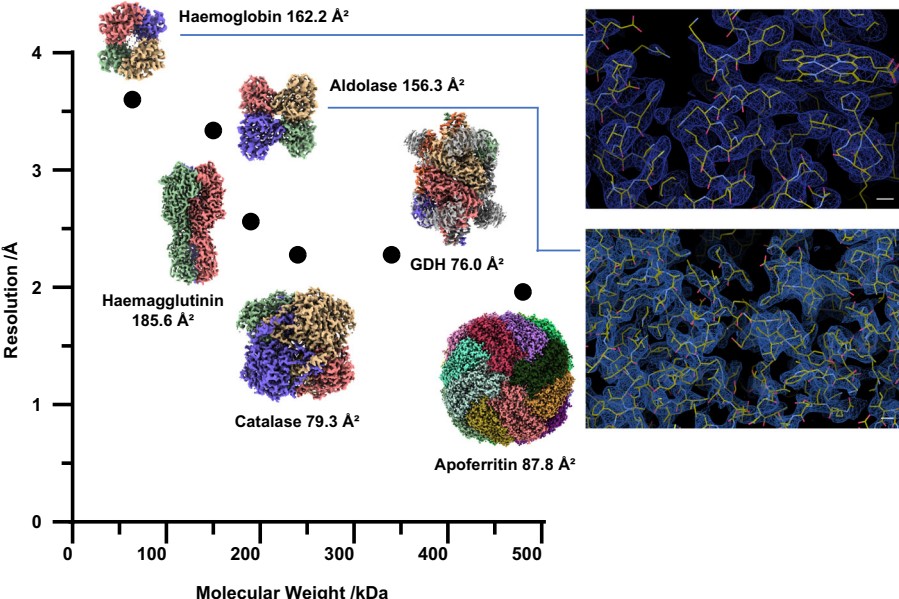

**Fig. 6 A plot showing the relationship between the molecular weight and the final resolution achieved in this study using the HFBI film.** Cryo-EM maps of haemoglobin (3.60 Å, 64 kDa), aldolase (3.28 Å, 150 kDa), influenza haemagglutinin trimer (2.56 Å, 190 kDa), catalase (2.29 Å, 240 kDa), GDH (2.26 Å, 334 kDa) and apoferritin (1.96 Å, 480 kDa) are shown with their Rosentha–Henderson *B*-factors. The representative regions of cryo-EM maps of haemoglobin and aldolase are magnified and shown on the right with their fitted atomic models. Scale bar, 2 Å.

**Application of HFBI film for small protein particles**. To further explore the potential of our HFBI film for cryo-EM experiments on small proteins (<200 kDa), we selected aldolase (150 kDa, homotetramer) and haemoglobin (64 kDa, heterotetramer αβαβ) for testing (Fig. 6).

We collected a large cryo-EM dataset of aldolase with the support of the HFBI film (Supplementary Table 1) and successfully completed single-particle analysis (Supplementary Fig. 15). We could visualize aldolase particles clearly from the raw micrographs (Supplementary Fig. 16a) and observe the secondary elements directly from the 2D classifications (Supplementary Fig. 16b). The cryo-EM map of aldolase was solved to a resolution of 3.28 Å (Fig. 6; Supplementary Fig. 16c) according to the gold standard threshold $FSC_{0.143}$ (Supplementary Fig. 16d). A nearly even orientation distribution was observed with the computed cryo-EF value of 0.84 (Supplementary Fig. 16e). The Rosenthal–Henderson *B*-factor was calculated as 156.3 Å² (Fig. 6; Supplementary Fig. 16f), which is twice as large as those in the above studies for apoferritin, catalase and GDH.

The increased Rosenthal–Henderson *B*-factor of the aldolase dataset could be attributed to the background of the HFBI film. Therefore, to process the cryo-EM dataset of haemoglobin with a molecular weight of 64 kDa, we began to consider the significant influence of this factor. The signal of the 2D crystal HFBI film could be removed using a lattice filtering algorithm in Fourier space[67,68] (Supplementary Fig. 17). Starting from the background extracted dataset, after several steps of single-particle analysis (Supplementary Fig. 18), we solved the cryo-EM map of haemoglobin at a resolution of 3.6 Å (Fig. 6; Supplementary Fig. 19a, b) according to the gold standard threshold $FSC_{0.143}$ (Supplementary Fig. 19c). No significant orientation bias was found with a cryo-EF value of 0.86 (Supplementary Fig. 19d, e). The Rosenthal–Henderson *B*-factor was calculated as 162.2 Å² (Fig. 6; Supplementary Fig. 19f).

We further explored whether the HFBI film could be suitable for even smaller proteins. Utilizing a similar approach to the previous study[17], starting from the raw dataset of 64 kDa haemoglobin particles, we performed in silica subtraction (Supplementary Fig. 20) to generate two datasets of smaller particles, one for a heterodimer (αβ subunit, 32 kDa) and another for a heterotrimer (αβα subunit, 48 kDa). Our subsequent image analysis showed that the 48 kDa heterotrimer could be resolved successfully to a high resolution of 3.8 Å, while the 32 kDa heterodimer could be resolved only to a low resolution of 6.4 Å.

## Discussion

In this study, we developed a new type of cryo-EM grid by coating amphipathic hydrophobin HFBI onto the holey ANTA foil grid and evaluated its ability to solve the air–water interface problem during the conventional vitrification procedure, which has been recognized as the major factor including denaturation/aggregation and preferred orientation of protein particles, preventing high-resolution results from cryo-EM.

The amphipathic property of the HFBI film allows it to easily coat the holey ANTA foil grid using its hydrophobic side and leave its hydrophilic side exposed to adsorb protein particles via electrostatic interaction. Thus, unlike other cryo-EM grids, pre-treatment such as plasma cleaning or UV irradiation before cryo-vitrification is not necessary. The adsorption of protein particles by the HFBI film not only protects the specimen from the air–water interface but also allows sufficiently thin ice to increase the quality of cryo-EM images. The quality of the cryo-EM dataset using the HFBI film was further proven by the small Rosenthal–Henderson *B*-factor in our experiments. With the HFBI film, we determined the high-resolution cryo-EM structures of catalase and influenza HA trimer, both of which had a severe preferential orientation problem using the conventional cryo-vitrification protocol.

Although the particle orientation distribution using the support of the HFBI film was not ideally uniform, the cryo-EF values for all our tested specimens were higher than 0.80, which was good enough to result in a sphericity of FSC close to 1.0. Thus, there was no strong preferential orientation problem when using the HFBI film support. In addition, we found that the HFBI film adsorbs protein particles via electrostatic interactions and that the particle orientation distribution can be regulated by adjusting the

pH. Based on this observation, we believe that for most specimens, to obtain well-distributed particles in cryo-EM experiments using the HFBI film, the optimal pH is close to the isoelectric point of the target complex.

The thickness of the HFBI film estimated by AFM is ~1.4 nm, which is thin enough to maintain a low background for most cryo-EM experiments. In this study, we proved that a standard image processing procedure was sufficient for particles with molecular weights higher than 150 kDa. With a lower molecular weight, the Rosenthal–Henderson $B$-factor increased, and the achievable resolution decreased (Fig. 6). For particles with molecular weights lower than 100 kDa, an additional background removal procedure would be important for high-resolution reconstruction. Since the HFBI film is a 2D crystal, its lattice background can be easily removed by the Fourier filtering algorithm without damaging the quality of the data. We proved that with this procedure, the structure of human haemoglobin (64 kDa) can be successfully solved to 3.6 Å. Our additional in silico experiments showed the potential application to molecular weights lower than 48 kDa.

Compared to other types of continuous support films, such as ultrathin carbon[13], graphene[14–20] or graphene derivates[21–28], lipid monolayers[29,30] and 2D crystals of streptavidin[31,32], our HFBI film approach is simple and efficient with a high coverage rate, does not require pre-treatment before cryo-vitrification, has a clear electrostatic mechanism of adhering particles, induces minimal contrast loss, and can yield thin enough ice for high-resolution data collection. Therefore, our HFBI film approach shows unique advantages to efficiently solve the air–water interface challenge and will have broad applications in future cryo-EM studies.

## Methods

**Expression and purification of hydrophobin HFBI**. The molecular cloning and recombinant expression of HFBI have been reported previously[52]. The genome cDNA of *T. reesei* strain (VTT D-98692) was generated by reverse transcription of total RNA. According to the HFBI sequence in Genbank (accession no. Z68124.1), gene *hfb*I was amplified by primers *Xho*I-*hfb*I-F (5′-CGCTCGAGAAAA-GAAGCAACGGCAACGGCAATG-3′) and *Eco*RI-*hfb*I-R (5′-CGGAATTCAAG-CACCGACGGCGGTCTG-3′) and was inserted into the pMD19-T vector. The *hfb*I gene in pMD19-T was then subcloned into the pPIC9 vector using both *Xho*I and *Eco*RI sites, yielding a recombinant plasmid pPIC9-*hfb*I, followed by transformation into *E. coli* strain DH5α. The plasmid pPIC9-*hfb*I was linearized with *Stu*I and transformed into competent *P. pastoris* GS115 His⁻ cells by electroporation using the Bio-Rad gene pulser apparatus (25 μF, 200 Ω, 2.0 kV). Positive transformants (His⁺ Mut⁺) were identified by colony PCR using primers AOX5 (5′-GACTGGTTCCAATTGACAAGC-3′) and AOX3 (5′-GCAAATGGCATTCT-GACATCC-3′). The selected positive transformants were cultured in buffered minimal glycerol (BMG) medium for bacterial body enrichment at 30 °C for 48 h and then induced in buffered minimal methanol (BMM) medium with 0.5% (v/v) methanol at 30 °C for 96 h to express recombinant HFBI. In addition, biotin was added into the medium when fermentation at a concentration of 0.4 mg l⁻¹.

HFBI was purified by a two-step procedure including ultrafiltration and acetonitrile extraction. The expression supernatant containing HFBI was collected and purified by ultrafiltration using a hollow fibre membrane module (Tianjin MOTIMO Membrane Technology Ltd., China) and then lyophilized to obtain a powder. The powder was redissolved in 0.01% trifluoroacetic acid/40% acetonitrile solution at a final concentration of 40 mg ml⁻¹ and stirred for 10 min prior to ultrasonication for 10 min. The crude extracted protein was collected by centrifugation at 8000 rpm for 30 min, lyophilized into powder and stored for further study.

**Water contact angle (WCA) measurements**. Hydrophobic siliconized glass (homemade) and hydrophilic mica sheets (J & K Chemical Technology, Tianjin, China) were used to study the properties of HFBI films on solid surfaces. Each surface was coated with 50 μl of 100 μg ml⁻¹ HFBI solution and incubated at room temperature (25 °C) overnight, using uncoated substrates as controls. The surface was subsequently rinsed three times with Milli-Q water to remove uncoated proteins and dried naturally. WCA was measured with a KSV contact angle measurement system with a 5 μl water droplet on the protein-coated side of the material. Three replicates were performed on different areas of the sample surface.

**Atomic force microscopy (AFM) measurements**. The freshly cleaved mica was coated with a 25 μl droplet of purified HFBI protein solution (10 μg ml⁻¹), incubated for 10 s at room temperature, and rinsed three times with Milli-Q water to remove uncoated protein. All samples were dried naturally and imaged immediately in the dry state using a Bruker Dimension FastScan Bio AFM ("Z" scanner; Digital Instruments/Veeco, USA) in ScanAsyst mode. The model of the probe was selected as FastScan-C, which matches the sharpness of a silicon tip with a low spring constant (0.8 N/m) and the high sensitivity of a silicon nitride cantilever. The AFM parameters were set as follows: scan size of 1.5 μm, scan rate of 1.95 Hz, resolution of 1024 × 1024 pixels, and aspect ratio of 1.0. The Limit of Z Range was 1 μm. The AFM images were processed using NanoScope analysis 1.80. Flattening of all acquired images was performed to remove possible tilt images. The thickness and roughness of the HFBI film were obtained by the Section and Roughness tool. Three different areas were analysed in the AFM images.

**Preparation of HFBI film-covered grid**. The purified and lyophilized powder of HFBI protein was dissolved in Milli-Q water (pH 7) at a concentration of 3.6 mg ml⁻¹, followed by sonication for ~1 min to prevent aggregation. Then, the solution was further diluted to a working concentration of 100 μg ml⁻¹, followed by additional sonication. A 30 μl drop of the working solution was placed on the surface of parafilm, which was placed in a cell culture dish and incubated for ~3 h in a humidified environment at room temperature and atmospheric pressure. When a visible film formed on the top of the drop, we transferred the film to a 300-mesh ANTA foil (1.2/1.3) Au grid (Zhenjiang Lehua Electronic Technology, China) by placing the ANTA foil side of the grid on top of the drop for 15 min (Fig. 2b). After the transfer, a 5 μl drop of Milli-Q water was added twice to wash the HFBI film, and filter paper was used to draw the remaining water away from the grid. Then, the grid was left to dry and stored in a grid box at room temperature.

**Preparation of graphene oxide-covered grid**. Graphene oxide sheets (Sigma-Aldrich 763705-25ML) were used to cover a 300-mesh ANTA foil (1.2/1.3) Au grid (Zhenjiang Lehua Electronic Technology, China) as described by Palovcak, E. et al.[22]

**Preparation of tested sample and cryo-vitrification**. Human apoferritin was provided by the laboratory of Prof. K.L. Fan and Prof. X.Y. Yan (Institute of Biophysics, CAS) and diluted to a concentration of 2 mg ml⁻¹ in buffer containing 20 mM Tris–HCl and 150 mM NaCl (pH 6.5). The influenza haemagglutinin trimer was purchased from MyBioSource (MBS434205) and used without dilution (1 mg ml⁻¹). Catalase from human erythrocytes, GDH, lyophilized rabbit muscle aldolase and lyophilized human haemoglobin were purchased from Sigma-Aldrich. Catalase was diluted to a final concentration of 2.3 mg ml⁻¹ in 50 mM Tris–HCl buffer (pH 6.5). Lyophilized rabbit muscle aldolase was solubilized to a final concentration of 1.2 mg ml⁻¹ in buffer containing 20 mM HEPES pH 7.5 and 50 mM NaCl. Lyophilized human haemoglobin was solubilized in PBS (phosphate-buffered saline, pH 7.5) buffer to a final concentration of 6.3 mg ml⁻¹. GDH was diluted to a final concentration of 3 mg ml⁻¹ in PBS buffers with the pH adjusted to 5.5, 6.0, 6.5, 7.0, 7.5 and 8.0, Tris buffer (20 mM Tris–HCl and 150 mM NaCl) with the pH adjusted to 8.5. All these specimens were used directly for subsequent cryo-vitrification without further treatments.

Using Vitrobot Mark IV (Thermo Fisher Scientific, USA), 3.0 μl aliquots of the sample were applied to the ANTA foil grid or the HFBI film-coated ANTA foil grid at 4 °C and 100% humidity. After waiting for 2 min, the grid was blotted for ~4 s and rapidly plunged into liquid ethane for cryo-vitrification.

**Cryo-EM data collection**. The cryo-EM dataset of apoferritin was collected using an FEI Talos Arctica electron microscope (200 kV) equipped with an energy filter and direct electron detector (Gatan BioQuantum K2) operated in superresolution mode (Supplementary Table 1). A total of 2649 raw movie stacks were automatically collected using SerialEM[69]. Images were recorded by the beam-image shift method[70] with a physical pixel size of 0.8 Å. The total dose was 50 e⁻/Å² to generate 32-frame gain normalized stacks in TIFF format. The defocus range varied from –0.6 to –2 μm.

Catalase datasets were collected using an FEI Titan Krios G2 electron microscope (300 kV) equipped with an energy filter and direct electron detector (Gatan BioQuantum K2) operated in superresolution mode (Supplementary Table 1). For the dataset using the ANTA foil grids, a total of 247 movies were automatically collected using SerialEM with a physical pixel size of 0.82 Å.

For the dataset using the HFBI film-coated ANTA foil grid, a total of 372 movies were automatically collected by SerialEM with a physical pixel size of 0.82 Å. To increase the resolution, another large dataset using the HFBI film-coated ANTA foil grid was collected with 2842 movies and a physical pixel size of 0.65 Å. Images were recorded by the beam-image shift method. The total dose was 60 e⁻/Å² to generate 40-frame gain normalized stacks in TIFF format. The defocus range varied from –0.6 to –1.6 μm.

The influenza haemagglutinin (HA) trimer dataset was collected in an FEI Talos Arctica electron microscope (200 kV) equipped with an energy filter and direct electron detector (Gatan BioQuantum K2) operated in superresolution mode (Supplementary Table 1). A total of 4574 raw movie stacks were automatically collected by SerialEM. Images were recorded by the beam-image shift method with

a physical pixel size of 0.80 Å. The total dose was 60 e⁻/Å² to generate 40-frame gain normalized stacks in TIFF format. The defocus range varied from –0.8 to –2.0 μm.

The aldolase dataset was collected using an FEI Titan Krios G2 electron microscope (300 kV) equipped with an energy filter and direct electron detector (Gatan BioQuantum K2) operated in superresolution mode (Supplementary Table 1). A total of 2460 raw movie stacks were automatically collected by SerialEM. Images were recorded by the beam-image shift method with a physical pixel size of 0.65 Å. The total dose was 70 e⁻/Å² to generate 40-frame gain normalized stacks in TIFF format. The defocus range varied from –0.6 to –1.6 μm.

The dataset of human haemoglobin was collected in an FEI Talos Arctica electron microscope (200 kV) equipped with an energy filter and direct electron detector (Gatan BioQuantum K2) operated in superresolution mode (Supplementary Table 1). A total of 2766 raw movie stacks were automatically collected by SerialEM. Images were recorded by the beam-image shift method with a physical pixel size of 0.65 Å. The total dose was 70 e⁻/Å² to generate 45-frame gain normalized stacks in TIFF format. The defocus range varied from –0.6 to –1.6 μm.

The datasets of GDH in a series of buffer conditions were collected using an FEI Titan Krios electron microscope (300 kV) equipped with a direct electron detector (Gatan BioQuantum K3) operated in superresolution mode. For specimens with pH values of 5.5, 6.0, 6.5, 7.0, 7.5, 8.0 and 8.5 vitrified using ANTA foil grids, 365, 321, 319, 335, 310, 299 and 304 movie stacks, respectively, were automatically collected by SerialEM. For specimens with pH values of 5.5, 6.0, 6.5, 7.0, 7.5, 8.0 and 8.5 vitrified using HFBI film-coated ANTA foil grids, 359, 294, 347, 349, 317, 313 and 776 movie stacks were automatically collected by SerialEM. Images were recorded by the beam-image shift method with a physical pixel size of 1.07 Å. The total dose was 60 e⁻/Å² to generate 32-frame gain normalized stacks in TIFF format. The defocus range varied from –0.6 to –1.6 μm.

Another large dataset of GDH buffered in PBS (pH = 7.5) was collected using an FEI Titan Krios G2 electron microscope (300 kV) equipped with an energy filter and direct electron detector (Gatan BioQuantum K2) operated in superresolution mode (Supplementary Table 1). A total of 1635 raw movie stacks were automatically collected by SerialEM. Images were recorded by the beam-image shift method with a physical pixel size of 0.65 Å. The total dose was 60 e⁻/Å² to generate 40-frame gain normalized stacks in TIFF format. The defocus range varied from –0.6 to –1.6 μm.

**Cryo-EM data processing**. For the small datasets of GDH collected under different buffer conditions and catalase using ANTA foil grids, all movie stacks were subjected to beam-induced motion correction and dose-weighting using MotionCorr2[71] with a 5 × 5 patch and a binning factor of 2. Contrast transfer function parameters for each micrograph were estimated by Gctf1.06[72]. Protein particles were picked by Gautomatch (developed by Kai Zhang, https://www2.mrc-lmb.cam.ac.uk/download/gautomatch-056/) using the same templates. The picked particles were imported into cryoSPARC[73] (v3.0), and multiple rounds of 2D classification were performed to discard contaminations and wrong particles.

For other large cryo-EM datasets, all movie stacks were subjected to beam-induced motion correction and dose-weighting using RELION3.1[74,75] with a 5 × 5 patch and a binning factor of 2. Contrast transfer function parameters for each micrograph were estimated by Gctf1.06[72]. Protein particles from a small subset of micrographs were picked by Gautomatch (developed by Kai Zhang, https://www2.mrc-lmb.cam.ac.uk/download/gautomatch-056/). These particles were subjected to multiple rounds of 2D classification to generate clean particles for the subsequent topaz training[76]. The topaz method was used for automatic particle picking based on deep-denoised micrographs[77]. In total, 351,565 apoferritin particles, 653,922 catalase particles, 1,213,983 influenza HA trimer particles, 35,587 GDH particles, 769,925 aldolase particles, and 874,836 haemoglobin particles were selected. Then, multiple subsequent rounds of 2D classification and 3D classification of binned particles in RELION3.1 were performed to discard "junk" particles.

For the apoferritin, catalase, GDH and aldolase datasets, 3D auto-refinement, CTF refinement and Bayesian polishing were all performed in RELION3.1[74,75] using the standard procedure. RELION's local resolution estimation was used to calculate the local resolution map. The final maps were post-processed in RELION3.1 for model building and refinement. Density modification of the apoferritin cryo-EM map was performed in PHENIX (v1.19-4092)[78] using the module of ResolveCryoEM[57].

For the influenza HA trimer dataset, the standard processing procedures, including 2D classification and 3D classification, were first performed in RELION3.1[74,75]. The particles were re-centred and re-extracted using the original pixel size of 0.8 Å. The subsequent heterogeneous refinement, nonuniform refinement with C3 symmetry, local CTF refinement and global CTF refinement were performed in cryoSPARC[73] (v3.0), and the pixel size was calibrated to 0.76 Å. The refined particles were then subjected to Bayesian polishing in RELION3.1. Finally, the polished particles were imported into cryoSPARC (v3.0) for final reconstruction with a resolution of 2.56 Å. Here, the transfer of parameters between cryoSPARC (v3.0) and RELION3.1 was performed using Pyem (v.0.5. Zenodo https://doi.org/10.5281/zenodo.3576630).

For the haemoglobin dataset, the standard processing procedures, including 2D classification and 3D classification, were first performed in RELION3.1[74,75]. Then, the background of the HFBI 2D crystal lattice was removed from the motion-corrected micrographs using the previously reported Fourier filtering script (https://doi.org/10.17632/k2g2p5z9x6.2)[67,68]. In brief, a median filter with a window size of 10 × 10 was first applied to the power spectrum of the micrograph to obtain the background with respect to the diffraction peaks of the HFBI 2D crystal lattice. This filtered spectrum was subsequently subtracted from the original power spectrum. After subtraction, the diffraction peaks could be easily identified using a proper threshold. Circle masks with a radius of 4 pixels were generated to cover these diffraction peaks and then applied to the Fourier transform of the original micrograph for subsequent Fourier filtering.

Starting from the Fourier-filtered micrographs, the particles were re-centred and re-extracted with the original pixel size of 0.65 Å. Using RELION3.1, the subsequent 3D auto-refinement with C2 symmetry and post-processing yielded a map with a global resolution of 4.25 Å. Then, the refined particles were subjected to an additional 3D classification in the local search mode. A total of 103,777 good particles were selected and subjected to 3D auto-refinement with enforced C2 symmetry. The global resolution was improved to 4.0 Å. The particles were then subjected to Bayesian polishing, improving the resolution to 3.9 Å. We then performed one round of random-phase 3D classification[79] to further remove bad particles. The final 58,828 good particles were subjected to further 3D auto-refinement and Bayesian polishing, improving the resolution to 3.69 Å. Finally, we imported the refined particles into M[80] and improved the final resolution to 3.6 Å for subsequent model building and refinement. Here, M[80] was also used to calculate the local resolution map.

**Model fitting and refinement**. The atomic models of apoferritin (PDB code 7K3V), catalase (PDB code 1DGH), influenza HA trimer (PDB code 6WXB), GDH (PDB code 3JCZ), aldolase (PDB code 6V20) and haemoglobin (PDB code 5NI1) were docked into the corresponding cryo-EM maps using UCSF-Chimera[81]. Then, each model was manually adjusted in Coot[82] and refined in real space using PHENIX[78] (Supplementary Table 1).

**Cryo-electron tomography data acquisition and processing**. We collected a cryo-electron tomography dataset of the apoferritin specimen vitrified using the HFBI film-coated ANTA foil grid. Tilt series were collected from −54° to 54° with an interval of 3° at 79,000 magnification (EFTEM mode, 1.7 Å pixel size) using an FEI Talos Arctica electron microscope (200 kV) equipped with an energy filter and direct electron detector (Gatan BioQuantum K2). For each tilt, the exposure time was set to 1.0 s with 8 frames using a total dose of 3.38 e⁻/Å² in electron counting mode. Each set of tilt series had a total dose of 125 e⁻/Å². Data acquisition was automatically performed using SerialEM[69]. The tilt series raw stacks were motion-corrected by MotionCor2[71]. After fiducial-free alignment using IMOD[83], the tomogram was reconstructed using ICON[84,85]. The pitch and off-set angles of the tomogram were determined using AutoGDeterm[86] and corrected using IMOD[83]. The final tomograms were generated with a binning factor of 4.

**Statistics and reproducibility**. All the representative electron micrographs in this work were repeated to acquire at least three times independently, which all resulted similar results. SDS–PAGE and AFM experiments were repeated independently at least three times with similar results.

**Reporting summary**. Further information on research design is available in the Nature Research Reporting Summary linked to this article.

## Data availability

The data that support this study are available from the corresponding authors upon reasonable request. The atomic coordinates and cryo-EM density maps of apoferritin, catalase, glutamate dehydrogenase, HA trimer, aldolase and haemoglobin have been deposited in the RCSB Protein Data Bank (PDB) with the accession codes 7VD8, 7VD9, 7VDA, 7VDF, 7VDC and 7VDE and in the Electron Microscopy Data Bank (EMDB) with the accession codes EMD-31910, EMD-31911, EMD-31912, EMD-31916, EMD-31913 and EMD-31915, respectively.

## Code availability

The script used for Fourier filtering is available at https://doi.org/10.17632/k2g2p5z9x6.2. The codes and scripts for electron tomography processing are available at the GitHub: https://github.com/hilbertsun/feilab/tree/main/tomography.

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

## Acknowledgements

We are grateful to Ping Shan, Ruigang Su and Mengyue Lou (F.S. lab) for their assistance in laboratory management. All sample preparation and cryo-EM work were performed at the Center for Biological Imaging (CBI, http://cbi.ibp.ac.cn), Institute of Biophysics, Chinese Academy of Sciences. We would like to thank Boling Zhu, Xiaojun Huang, Xujing Li, Tongxin Niu, Shuangbo Zhang, and Lihong Chen from CBI for their help with cryo-EM data collection, Deyin Fan from CBI for his help with the preparation of ultrathin carbon film, Xin Jia from CBI for her help with the SEM experiments, and Cui Song from State Key Laboratory of Biochemical Engineering, Institute of Process Engineering, Chinese Academy of Science for her help with the AFM experiments. This work was equally supported by grants from the Ministry of Science and Technology of China (2017YFA0504700 to F.S.), the National Natural Science Foundation of China (31830020 to F.S.), and the Chinese Academy of Sciences (XDB37040102 to F.S.). The authors are also grateful for the grant support from the National Natural Science Foundation of China for Distinguished Young Scholars (31925026 to F.S.), from the Beijing Municipal Science and Technology Commission (Z181100004218002 to F.S.) and from the Sino-Swiss Scientific and Technological Cooperation Project by the Ministry of Science and Technology of China (2015DFG32140 to M.Q.).

## Author contributions

F.S. initiated the project. F.S. and M.Q. supervised the project. H.F. designed and performed the production of HFBI film-covered grids. B.W., B.S. and H.X. expressed, purified and characterized HFBI proteins. H.F. performed all cryo-EM sample preparation, data collection, imaging processing and model building. H.F., F.S., Yan Z., Yun Z. and Yujia Z. analysed the data. H.F. and F.S. wrote the manuscript with additional input from B.W., H.X. and M.Q.

## Competing interests

Parts of this study (the production of HFBI film-covered grids) have been submitted as a Chinese patent of invention (belonging to H.F., F.S., B.W., M.Q., and Yun Z.) with the application number 202110576212.4 and are currently under scrutiny. The remaining authors declare no competing interests.
