## [Peer Review File · Nature Communications]

A cryo-electron microscopy support film formed by 2D crystals
of hydrophobin HFBIEditorial Note: This manuscript has been previously reviewed at another journal that is not operating a transparent peer review scheme. This document only contains reviewer comments and rebuttal letters for versions considered at *Nature Communications*.

REVIEWERS' COMMENTS

Reviewer #2 (Remarks to the Author):

The authors have taken a great effort to address all the reviewers' comments and the manuscript is greatly improved. This work is now ready to be published in a timely manner.

Reviewer #3 (Remarks to the Author):

The manuscript was extensively revised and improved significantly. Notably, the figures are enhanced considerably. In addition, the additional experiments mostly clarified my concerns about the electrostatic interaction of the particles with the new interface.

The additional experiments indeed show the protein adsorption's pH (electrostatic) dependency (supplemental Figure 11). But it does not explain the "worm-like" structures. In my view, this indicates that the interaction force between the particles is not obscured anymore by other forces, e.g., the interaction with the interface with the particles enforcing top views. Furthermore, unfortunately, manuscript Fig. 5a provides only relative information. It would be essential to know the absolute number of particles picked, discarded, and used for the reconstruction. And, finally, how many of the particles were integrated into the "worms."

I believe the proposed methodology should be shared with the community asap and deserves publication in nature communications.

Minor comments

- Abstract: "air-water interface-induced": Missing coma, or should second hyphenation be removed? (I'm not a native English speaker)

- Supplementary Fig 4: The contamination is also indicated by a cross, which is not described in the legend.

Letter to response reviewers

We would like to appreciate the original reviewers to check our revision again and highly recommend our work. According to the minor comments of Reviewer #3, we made a final revision of our manuscript with the following editions.

1. Providing the absolute numbers of particles picked and used for the reconstruction in Fig. 5a.
2. Adding more discussions about the formation of “worm-like” structure of GDH.
3. Corrections of typos indicated by the reviewer.

The point-by-point responses to the reviewers’ suggestions and comments are given below.

[To Reviewer #2]

Remarks to the Author:

The authors have taken a great effort to address all the reviewers' comments and the manuscript is greatly improved. This work is now ready to be published in a timely manner.

Response #1: We appreciate this reviewer for his/her high evaluation of our work. We believe our work will make a great contribution to the cryoEM community.

[To Reviewer #3]

Remarks to the Author:

The manuscript was extensively revised and improved significantly. Notably, the figures are enhanced considerably. In addition, the additional experiments mostly clarified my concerns about the electrostatic interaction of the particles with the new interface.

Response #1: We are very happy to know our revision has satisfied this reviewer.

The additional experiments indeed show the protein adsorption's pH (electrostatic) dependency (supplemental Figure 11). But it does not explain the "worm-like" structures. In my view, this indicates that the interaction force between the particles is not obscured anymore by other forces, e.g., the interaction with the interface with the particles enforcing top views. Furthermore, unfortunately, manuscript Fig. 5a provides only relative information. It would be essential to know the absolute number of particles picked, discarded, and used for the reconstruction. And, finally, how many of the particles were integrated into the "worms."

Response #2: Thanks a lot for this suggestion.

For the “worm-like” structure, we agree with this reviewer that it indicates a significant top-to-top interaction between GDH particles, which is the intrinsic nature of GDH protein. At the condition without HFBI film, the interaction between air-water interface (AWI) and the top side of GDH is stronger than the top-to-top interaction itself, which enforces the majority of GDH top views. At the condition with HFBI film, the interaction between the new interface and the top side of GDH is regulated efficiently by pH and becomes weaker than the top-to-top interaction of GDH when pH is larger than 7.0, which allows significant population of side views. Indeed, the “worm-like” structures were also observed from the recent report when the property of AWI is changed by adding amount of cationic detergent (see Figure S7 in PMID 34454014). In our observations, there are a large portion of side view particles integrated into the “worms”, which is also consistent with this report (PMID 34454014). These discussions have been added into the revise text as follows.

“The “worm-like” structure of GDH (Supplementary Fig. 11a-g) suggests an intrinsic top-to-top interaction between GDH particles. At the condition without HFBI film, the interaction between air-water interface and the top side of GDH is stronger than the top-to-top interaction itself, which enforces the majority of GDH top views. At the condition with HFBI film, the interaction between the new interface and the top side

of GDH is regulated efficiently by pH and becomes weaker than the top-to-top interaction of GDH when pH is larger than 7.0, which allows significant population of side views. Indeed, the “worm-like” structures were also observed from the recent report when the property of air-water interface is changed by adding amount of cationic detergent.”

The absolute numbers of particles picked and used for the reconstruction in Fig. 5a have been provided in the revised text as follows.

“We found that the orientation distributions of GDH changed at different pH conditions from 5.5 to 8.5 (Fig. 5a; Supplementary Fig. 11a-g). For specimens with pH values of 5.5, 6.0, 6.5, 7.0, 7.5, 8.0 and 8.5 vitrified using HFBI film-coated ANTA foil grids, 240,067, 198,712, 228,000, 241,074, 223,500, 216,510 and 527,163 particles were automatically picked. Multiple rounds of 2D classification were performed to discard contaminations and wrong particles. Final 162,042 (48,499 for side views), 120,274 (14,013 for side views), 149,320 (37,812 for side views), 120,149 (73,256 for side views), 82,340 (52,545 for side views), 88,585 (60,808 for side views), 174,836 (126,218 for side views) particles were selected. ...”

“To exclude the possibility that our observed pH-dependent change in orientation distribution might be induced by pH-dependent characteristics of GDH itself, we performed control cryo-EM experiments using normal ANTA foil grids (Fig. 5a; Supplementary Fig. 11a-g). For specimens with pH values of 5.5, 6.0, 6.5, 7.0, 7.5, 8.0 and 8.5 vitrified using ANTA foil grids, 300,185, 255,171, 243,678, 265,459, 257,553, 235,488 and 267,823 particles were automatically picked. Final 183,707 (71,119 for side views), 137,079 (32,461 for side views), 96,023 (15,064 for side views), 138,633 (43,985 for side views), 169,991 (34,426 for side views), 150,087 (34,041 for side views), 239,545 (73,569 for side views) particles were selected. ...”

I believe the proposed methodology should be shared with the community asap and deserves publication in nature communications.

Response #3: We appreciate this reviewer for the high evaluation of our work. We believe our work will make a great contribution to the cryoEM community.

Minor comments

- Abstract: "air-water interface-induced": Missing coma, or should second hyphenation be removed? (I'm not a native English speaker)

Response #4: We have removed the second hyphenation.

- Supplementary Fig 4: The contamination is also indicated by a cross, which is not described in the legend.

Response #5: The yellow cross does not indicate the contamination but the position of ice at AWI. It was generated in IMOD and left in the snapshot. This information was added in the revised legend. We also added a description of green lines in the legend.